# FROM MODEL FUSION TO LAYER FUSION: A WEIGHT PERSPECTIVE

## ABSTRACT

Although large language models (LLMs) have demonstrated remarkable performance in natural language processing tasks, their **massive parameter counts** and **high inference costs** severely limit practical applications. Existing lightweight approaches, such as quantization, knowledge distillation, and pruning, often suffer from significant **performance degradation**, heavy **reliance on fine-tuning**, or **insufficient hardware support**. In recent years, layer pruning has gained attention as a structurally friendly compression strategy. However, existing methods still **struggle to adequately preserve the functional information** within removed layers and typically **require complex post-processing**. To address these issues, we propose a novel Layer Fusion (LF) framework, which compresses models by fusing functional weights across multiple Transformer layers with **no fine-tuning required** and **without extensive data requirements**. The LF framework consists of five core modules: identifying layer features, determining fusion targets, extracting residual weights, balancing parameter importance, and generating composite weights through fusion. This approach requires only **a small amount of probe data** and **facilitates efficient hardware inference**. Experiments demonstrate that LF significantly **outperforms mainstream model compression techniques** across multiple benchmarks and model architectures, achieving a superior performance-size trade-off with lower computational overhead. Moreover, LF exhibits **strong scalability and compatibility**, offering a new direction for model compression research. Our code has been released on the anonymous github.

## 1 INTRODUCTION

Traditional LLM (large language model) compression methods primarily include quantization Zhou et al. (2024), knowledge distillation Gou et al. (2021), and pruning Liu et al. (2018). Quantization reduces the numerical precision during inference (e.g., converting 32-bit floating-point numbers to 16-bit) to compress the model. Empirical results show that this method has a minimal impact on performance within certain accuracy ranges and can be easily combined with other compression techniques Egashira et al. (2024). Knowledge distillation utilizes a larger, more powerful teacher model to generate high-quality annotations for training a lightweight student model, aiming to approximate the performance of the original model. However, this approach typically demands substantial computational resources and training time, posing practical barriers Cho & Hariharan (2019). Pruning can be categorized into unstructured and structured pruning: unstructured pruning Liao et al. (2023); Bowen et al. (2024) identifies redundant weights through importance evaluation (e.g., magnitude or Taylor expansion estimates) and sets them to zero, but the resulting sparse matrices are often difficult to accelerate efficiently on hardware; structured pruning Fang et al. (2023) alleviates this issue to some extent by removing entire rows or columns of weights, yet it often leads to irregular model architectures, limiting flexible deployment.

Recently, a new compression method—layer pruning—has garnered increasing attention. Based on the assumption of redundancy among Transformer layers Gromov et al. (2024), this approach directly removes certain layers while striving to preserve model performance. Since removing entire layers does not alter the computational graph structure, it naturally supports hardware acceleration and exhibits considerable application potential. Existing studies can be divided into two categories: one directly identifies and eliminates redundant layers Song et al. (2024) Men et al. (2024), while the other involves fine-tuning after pruning to recover performance Kim et al. (2024) Gromov et al.

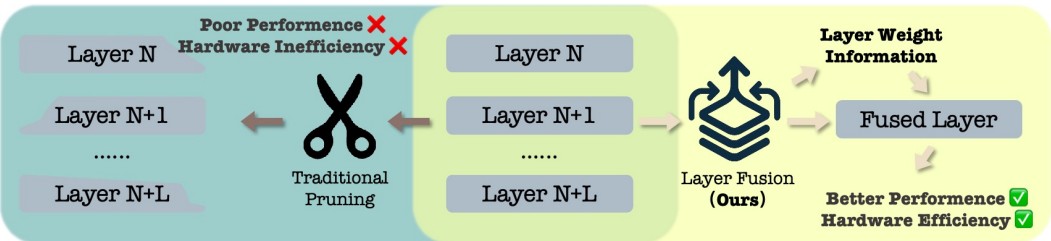

Figure 1: Our layer fusion approach differs from traditional pruning.

(2024) Chen et al. (2024). However, directly removing layers often results in significant performance degradation, and fine-tuning requires extensive data and training resources. Although some studies have attempted to reduce fine-tuning costs Chen et al. (2024), they still necessitate tens of thousands of data samples, limiting practical applicability. Notably, these methods generally overlook the underutilized functional information within the pruned layers.

To better leverage the functional weights of the pruned layers, we propose a weight fusion approach to retain the functionalities of the original layers. Language models typically consist of stacked Transformer decoder layers with homogeneous structures, a characteristic highly similar to the settings of multi-task model weight fusion Ilharco et al. (2022); Ainsworth et al. (2022). Therefore, each layer can be regarded as a functional sub-model. By fusing the weights of multiple layers, a multifunctional composite layer is formed, thereby preserving performance while compressing.

Motivated by this and inspired by key work in the field of weight fusion Ilharco et al. (2022), we propose a novel layer fusion (**LF**) framework, which comprises five core modules: Identification, Decision, Residual Extraction, Balancing, and Fusion. Specifically:

**1) Identification**: A small amount of probe data ($\leq 50$ samples) is fed into the model to extract the input and output hidden states of each decoder layer. **2) Decision**: Based on a user-specified compression ratio and layer interval $N$, the similarity of input/output states across consecutive $N$ layers is computed. The consecutive layers with the highest similarity (i.e., the greatest functional overlap) are selected for fusion. **3) Residual Extraction**: The functional weighted average of the selected $N$ layer weights are computed as the baseline weights section 3. The residual between each layer's weights and this baseline is calculated to obtain a layer vector (**LV**) representing layer-specific information. **4) Balancing**: To reduce redundancy and noise, importance weighting is applied to each dimension of the layer vector, emphasizing information-rich parameters. **5) Fusion**: The weighted layer vectors are fused with the baseline weights to generate a new weight matrix, replacing the original $N$ decoder layers.

Our method requires only minimal data and achieves high-performance compression without fine-tuning, offering computational efficiency and hardware-friendliness, making it suitable for plug-and-play model compression scenarios. Additionally, the LF framework is highly modular and extensible, with each stage allowing independent optimization.

**The main contributions of this paper are as follows:** **i)** We pioneer the concept of **layer fusion** from a **weight fusion** perspective and systematically analyze its similarities and differences with related methods, providing new insights for future research. **ii)** The proposed layer fusion framework exhibits strong **compatibility** and **extensibility**, enabling integration with existing weight fusion techniques and advancing the field of model compression. **iii)** Our approach significantly outperforms mainstream baseline methods with almost no additional computational overhead, enhancing the practical value of lightweight technologies. **iv)** We have made our implementation code **publicly available**. We believe this will promote the development of the community.

**Most Relevant Work Yang et al. (2024b)** This paper introduces a method called LaCo, which fuses $N$ consecutive layers of a model into a single layer. The method selects the first layer as the baseline, calculates the difference between the weights of subsequent layers and the first layer, and then directly adds these differences to the weights of the first layer to obtain the fused layer. However, based on our observations, directly adding the differences to the baseline layer results in excessively large weights, severely impacting performance. Furthermore, LaCo does not address the relationship between its method and model fusion, nor does it provide a more granular analysis or explanation of this fusion process. These aspects are comprehensively covered in our paper.

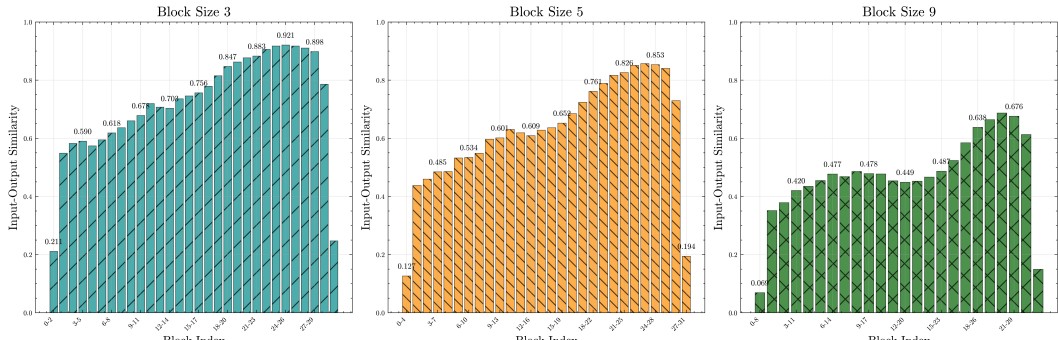

Figure 2: We partitioned the layers of Llama 3.1-8B Dubey et al. (2024) according to different block sizes and analyzed the similarity between the hidden states of each block's inputs and outputs. As shown in the figure, the **similarity** between block inputs and outputs **increases** at deeper layers, partially demonstrating redundancy characteristics. This provides intuitive support for the rationale behind our layer fusion approach.

## 2 PRELIMINARY

In this section, we will provide a detailed introduction to **fundamental model fusion** techniques and the **rigorous definition of model layer compression**. This lays the groundwork for subsequent discussions of our approach.

### 2.1 MODEL FUSION: TASK VECTOR BASED

Recent work by Ilharco et al. (2022) introduced task vectors as a mechanism for steering the behavior of pre-trained models through arithmetic operations in weight space. Formally, given a pre-trained model with parameters $\theta_{\text{pre}} \in \mathbb{R}^d$ and a model fine-tuned on a task $t$ with parameters $\theta_t^{\text{ft}} \in \mathbb{R}^d$, the task vector $\tau_t$ is defined as the element-wise difference:

$$\tau_t = \theta_t^{\text{ft}} - \theta_{\text{pre}} \tag{1}$$

This vector encodes the direction in weight space that improves performance on task $t$. Task vectors can be scaled and **combined through arithmetic operations** to edit model behavior without additional training.

If we want to integrate $N$ downstream task models that perform different tasks, we can fuse the task vectors corresponding to these downstream task models and integrating them onto the pre-trained weights.

$$\theta^{\text{fused}} = \theta_{\text{pre}} + \lambda \sum_{t=1}^{N} \theta_t^{\text{ft}} \tag{2}$$

The resulting fusion model $\theta^{\text{fused}}$ possesses the capabilities of the N downstream models that were fused. This idea has inspired us to explore **the integration of layers within the model**.

### 2.2 PROBLEM FORMULATION: RIGOROUS DEFINITION OF MODEL LAYER COMPRESSION

Let a deep neural network model be represented as a sequential composition of $L$ layers. Formally, the model $\mathcal{M}$ is defined as:

$$\mathcal{M} = \mathcal{L}_L \circ \mathcal{L}_{L-1} \circ \cdots \circ \mathcal{L}_1 \tag{3}$$

where each layer $\mathcal{L}_i$ for $i = 1, 2, \ldots, L$ is a parametric function with parameters $\theta_i \in \mathbb{R}^{d_i}$, and $\circ$ denotes function composition. The overall parameter set of the model is $\Theta = \{\theta_1, \theta_2, \ldots, \theta_L\}$. Given an input $\mathbf{x}$, the output is computed as $\mathbf{y} = \mathcal{M}(\mathbf{x}; \Theta)$.

**Layer Redundancy Hypothesis** The theoretical foundation of layer compression rests on the layer redundancy hypothesis, which posits that deep neural networks inherently contain significant functional redundancy across adjacent layers Figure 2. Formally, for a sequence of consecutive layers

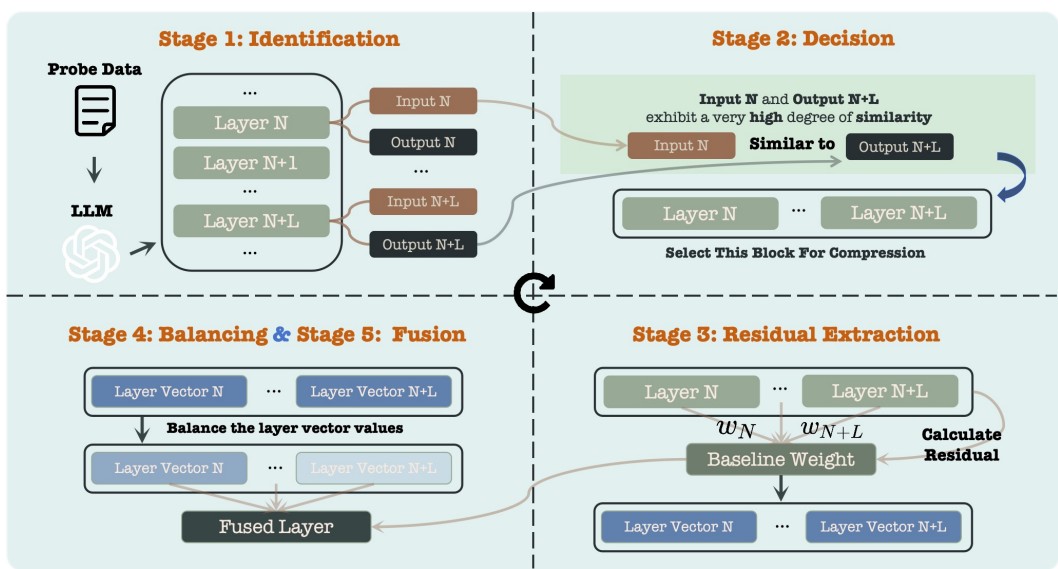

Figure 3: A brief summary of our layer fusion (**LF**) framework, the five stages detailed in section 3.

$\{\mathcal{L}_i, \mathcal{L}_{i+1}, \ldots, \mathcal{L}_{i+k-1}\}$, we hypothesize that their composed transformation can be sufficiently approximated by a more compact representation:

$$\mathcal{L}_{i+k-1} \circ \cdots \circ \mathcal{L}_{i+1} \circ \mathcal{L}_i(\mathbf{x}) \approx \tilde{\mathcal{L}}(\mathbf{x}) \tag{4}$$

where $\tilde{\mathcal{L}}$ denotes a compressed transformation that preserves the essential functionality of the original $k$ layers. This hypothesis suggests that the parameter spaces of adjacent layers exhibit substantial linear dependence and functional similarity, creating opportunities for depth reduction without significant performance degradation.

The existence of such layer redundancy provides both the motivation and theoretical justification for model layer compression, indicating that careful recombination of layer parameters can maintain network performance while substantially reducing computational requirements.

**Model Layer Compression** aims to reduce the number of layers while preserving the model's performance. Specifically, we seek a compressed model $\mathcal{M}'$ with $L'$ layers where $L' < L$:

$$\mathcal{M}' = \mathcal{L}'_{L'} \circ \mathcal{L}'_{L'-1} \circ \cdots \circ \mathcal{L}'_1 \tag{5}$$

with parameters $\Theta' = \{\theta'_1, \theta'_2, \ldots, \theta'_{L'}\}$. The goal is to ensure that the behavior of $\mathcal{M}'$ approximates that of $\mathcal{M}$ over an input distribution $\mathcal{X}$. This is formalized by minimizing a performance gap:

$$\min_{\Theta'} \mathbb{E}_{\mathbf{x} \sim \mathcal{X}} \left[ \mathcal{D} \left( \mathcal{M}(\mathbf{x}; \Theta), \mathcal{M}'(\mathbf{x}; \Theta') \right) \right] \tag{6}$$

where $\mathcal{D}$ is an appropriate discrepancy measure between outputs, and the compression ratio is $\rho = (L - L')/L$.

This compression process involves identifying a transformation $f : \Theta \to \Theta'$ that **reduces layer count** while **preserving functional behavior**, leveraging the inherent redundancies identified in the layer redundancy hypothesis.

## 3 LAYER FUSION: OUR METHOD

### 3.1 OVERVIEW

**Layer Fusion (LF)** is a structured compression framework designed to reduce the depth of LLM by fusing contiguous layers with low functional strength subsection 3.2. The method operates through five sequential stages: (1) **Identification**, where layer-wise activations are recorded using probe data; (2) **Decision**, where fusion blocks are selected based on activation similarity and user-defined compression ratio; (3) **Residual Extraction**, which computes layer-specific residuals relative to

a geometric centroid; (4) **Balancing**, which emphasizes salient features in the residuals; and (5) **Fusion**, where residuals are combined and merged into a new layer. An overview of the pipeline is illustrated in Figure 3.

## 3.2 FUNCTIONAL STRENGTH METRIC

Before introducing our method, we first define a key metric, which we refer to as **functional strength**. Suppose we now have a continuous network layer represented as $S = \{\mathcal{L}_t, \mathcal{L}_{t+1}, \ldots, \mathcal{L}_{t+B-1}\}$. Here, $S$ denotes a continuous layer block. Meanwhile, $B$ indicates the length of this contiguous block. We can define the input-output similarity of this continuous block as:

$$\text{Sim}(S) = \frac{1}{B} \sum_{n=1}^{B} \kappa \left( \mathbf{h}_{t-1}^{(n)}, \mathbf{h}_{t+B-1}^{(n)} \right) \tag{7}$$

Where $\mathbf{h}_i$ denotes the hidden state output of the i-th layer of the model, as detailed in subsection 3.3, and $\kappa(\cdot, \cdot)$ denotes a similarity function (In our implementation, we employ the most commonly used cosine similarity). Based on the assumption, blocks with high input-output similarity exhibit lower functionality (exerting minimal influence on hidden states). We can define the functional strength of a block as:

$$\text{Func}(S) = 1 - \text{Sim}(S) \tag{8}$$

We can use this method to calculate the functional strength of a block in the subsequent discussion, and it can also be applied to calculate the functional strength of a single layer ($B = 1$).

## 3.3 STAGE 1: IDENTIFICATION

To provide essential reference information for subsequent fusion compression processes, we constructed a set of probe data to collect the hidden states of each layer in the model.

Let the original model $\mathcal{M}$ consist of $L$ layers. We sample a small set of probe data $\mathcal{P} = \{\mathbf{x}_1, \mathbf{x}_2, \ldots, \mathbf{x}_N\}$, where $N \leq 50$, covering diverse input types. For each input $\mathbf{x} \in \mathcal{P}$, we record the hidden state after each layer:

$$\mathbf{h}_i = \mathcal{L}_i(\mathbf{h}_{i-1}), \quad \text{for } i = 1, 2, \ldots, L \tag{9}$$

with $\mathbf{h}_0 = \mathbf{x}$. The collected hidden states form a matrix $\mathbf{H} \in \mathbb{R}^{L \times N \times d}$, where $d$ is the hidden dimension.

## 3.4 STAGE 2: DECISION

Unlike many other approaches, we consider more than just compressing a single contiguous block. We support users setting different block sizes and compressing multiple blocks discretely to achieve a specified compression ratio. This method offers greater flexibility and control. The traditional approach focusing solely on a single block can be viewed as a **special case** of our method.

Given a target compression ratio $\rho < 1$ and a block size $B \in \mathbb{Z}^+$, we aim to reduce the number of layers such that:

$$L' = \lfloor L \cdot (1 - \rho) \rfloor \tag{10}$$

We generate all contiguous blocks of $B$ layers, denoted as $\mathcal{B} = \{S_j\}_{j=1}^{L-B+1}$, where $S_j = \{\mathcal{L}_j, \mathcal{L}_{j+1}, \ldots, \mathcal{L}_{j+B-1}\}$. For each block $S_j$, we compute the functional strength $\text{Func}(S_j)$. And select non-overlapping blocks $\{S_{j_1}, S_{j_2}, \ldots, S_{j_K}\}$ that **minimize** the total functional strength to ensure that the blocks we prepare for compression have minimal impact on the overall performance:

$$\min \sum_{k=1}^{K} \text{Func}(S_{j_k}), \quad \text{subject to } K \cdot (B - 1) = L - L' \tag{11}$$

ensuring the total number of layers removed aligns with the compression ratio.

## 3.5 STAGE 3: RESIDUAL EXTRACTION

For each selected block $S = \{\mathcal{L}_t, \mathcal{L}_{t+1}, \ldots, \mathcal{L}_{t+B-1}\}$ with parameters $\{\theta_t, \theta_{t+1}, \ldots, \theta_{t+B-1}\}$, we compute the baseline weight:

$$\bar{\theta} = \text{Centroid}(\theta_t, \theta_{t+1}, \ldots, \theta_{t+B-1}) \tag{12}$$

The layer vector (residual) for each layer $\mathcal{L}_i$ is defined as:

$$\mathbf{v}_i = \theta_i - \bar{\theta}, \quad \text{for } i = t, t+1, \ldots, t+B-1 \tag{13}$$

These vectors capture layer-specific deviations from the centroid. We have experimentally and theoretically demonstrated that layer vector fusion exhibits **superior orthogonality** compared to direct fusion of layer parameters.

**Extension** We primarily employed the Functional Weighted Average method for selecting the centroid. Weights are assigned based on the functional strength indicators of each layer within the current compressed block, yielding a weighted average to obtain the baseline weight:

$$\text{Centroid}(\theta_t, \theta_{t+1}, \ldots, \theta_{t+B-1}) = \sum_{i=t}^{t+B-1} \frac{\text{Func}(\mathcal{L}_i)}{\sum_{j=t}^{t+B-1} \text{Func}(\mathcal{L}_j)} \theta_i \tag{14}$$

We theoretically evaluate the advantages and disadvantages of three distinct baseline weighting methods and conclude that the **functional weighted average (Ours)** approach provides a superior foundation for layer vector extraction. The proof process is detailed in Appendix C and Appendix D.

### 3.6 STAGE 4: BALANCING

To emphasize important features in the layer vectors, we apply a balancing mechanism $\psi : \mathbb{R}^d \to \mathbb{R}^d$ that scales each dimension of $\mathbf{v}_i$ based on its magnitude:

$$\tilde{\mathbf{v}}_i = \psi(\mathbf{v}_i) = \mathbf{v}_i \odot \mathbf{w}_i \tag{15}$$

where $\mathbf{w}_i \in \mathbb{R}^d$ is a weight vector whose components are monotonically increasing functions of $|v_{i,k}|$. This suppresses noise while preserving salient features.

In our actual implementation, we applied the balancing method described in Du et al. (2024). Further details will be provided in Appendix E.

### 3.7 STAGE 5: FUSION

The balanced residuals are fused into a single residual vector:

$$\tilde{\mathbf{v}} = \lambda \sum_{i=t}^{t+B-1} \tilde{\mathbf{v}}_i \tag{16}$$

Here, $\lambda$ represents the fusion coefficient, which controls the overall amplitude of the layer vector during fusion. $\tilde{\mathbf{v}}$ is added to the centroid to produce the fused layer parameters:

$$\theta_{\text{fused}} = \bar{\theta} + \tilde{\mathbf{v}} \tag{17}$$

The new layer $\mathcal{L}_{\text{fused}}$ with parameters $\theta_{\text{fused}}$ replaces the original $B$ layers. The process is **repeated for all selected blocks**, resulting in a compressed model $\mathcal{M}'$ with $L'$ layers.

**Discussion: What does the layer fusion framework bring?** Traditional unstructured pruning methods and model fusion approaches face significant limitations in practical application. Unstructured pruning is difficult to leverage directly due to the challenges of hardware acceleration Yang & Zhang (2021), while model fusion often suffers from substantial performance degradation Yang et al. (2024a), hindering its deployment. However, our **layer fusion** framework effectively **combines these two techniques** for more promising and practical model compression tasks, unlocking greater potential for future advancements in these fields.

## 4 EXPERIMENTS

### 4.1 BENCHMARK

**Dataset** To comprehensively evaluate the performance of compressed models, we selected six authoritative benchmark datasets: **ARC (Easy & Challenge)** Clark et al. (2018), **HellaSwag** Zellers et al. (2019), **OpenBookQA** Mihaylov et al. (2018), **PIQA** Bisk et al. (2020), and **Winogrande ai2 (2019)**. This combination spans multiple cognitive dimensions—including scientific knowledge, common-sense reasoning, physical understanding, and linguistic disambiguation—effectively

Table 1: Performance Comparison on Multiple Datasets under Different Compression Ratios. Dense models serve as original model. Best results for each dataset are highlighted in **bold**, and second-best results are _underlined_. CR represents compression ratio ($\rho$).

| CR | LLM | Method | Dataset Performance (%) | | | | | | |
|---|---|---|---|---|---|---|---|---|---|
| | | | ARC-c | ARC-e | HellaSwag | OpenBookQA | PIQA | WinoGrande | Average |
| 25% | Llama3.1-8B | _Dense_ | 55.29 | 79.67 | 79.21 | 43.20 | 80.97 | 74.03 | 68.73 |
| | | LLMPruner | 25.94 | 26.22 | 26.05 | 26.40 | 50.82 | 48.78 | 34.04 |
| | | SliceGPT | 20.48 | 33.88 | 28.34 | 26.00 | 52.83 | 50.83 | 35.39 |
| | | LaCo | 29.92 | 27.39 | 27.42 | 30.40 | 53.32 | 52.25 | 36.78 |
| | | LF (Avg) | 38.57 | _47.73_ | 56.35 | _31.80_ | 68.28 | 65.59 | _51.39_ |
| | | LF (First) | _38.74_ | 46.12 | **58.92** | 31.40 | 67.28 | **65.87** | _51.39_ |
| | | LF (Centroid) | **38.82** | **47.94** | _56.50_ | **32.00** | 68.82 | 65.43 | **51.59** |
| | Llama2-13B | _Dense_ | 49.32 | 77.53 | 79.40 | 45.20 | 80.47 | 71.74 | 67.28 |
| | | LLMPruner | 34.39 | 62.75 | 52.46 | 36.00 | 72.47 | 53.59 | 51.94 |
| | | SliceGPT | 38.23 | 60.94 | 57.39 | 40.60 | 67.08 | 67.88 | 55.35 |
| | | LaCo | 40.27 | 59.34 | 66.00 | 36.80 | 70.57 | 69.38 | 57.06 |
| | | LF (Avg) | 41.72 | 61.91 | 67.29 | 36.60 | 71.20 | _69.22_ | 57.99 |
| | | LF (First) | _41.81_ | _61.99_ | _67.32_ | _37.00_ | _71.44_ | 68.98 | _58.09_ |
| | | LF (Centroid) | **42.66** | **62.25** | **67.93** | **37.40** | **71.65** | **70.17** | **58.68** |
| 12.5% | Llama3.1-8B | _Dense_ | 55.29 | 79.67 | 79.21 | 43.20 | 80.97 | 74.03 | 68.73 |
| | | LLMPruner | 27.22 | 26.26 | 26.53 | 26.60 | 50.98 | 49.17 | 34.46 |
| | | SliceGPT | 20.90 | 36.83 | 29.75 | 25.80 | 54.68 | 49.72 | 36.28 |
| | | LaCo | 47.03 | 67.63 | 70.51 | 38.20 | 74.27 | 71.19 | 61.47 |
| | | LF (Avg) | _48.29_ | _67.80_ | _70.01_ | _40.80_ | _73.39_ | 70.64 | _61.82_ |
| | | LF (First) | 46.67 | 66.46 | 68.84 | **41.00** | 73.07 | **70.96** | 61.17 |
| | | LF (Centroid) | **48.63** | **68.43** | **70.37** | **41.00** | **73.94** | _70.72_ | **62.18** |
| | Llama2-13B | _Dense_ | 49.32 | 77.53 | 79.40 | 45.20 | 80.47 | 71.74 | 67.28 |
| | | LLMPruner | 45.14 | 73.36 | 72.99 | 41.40 | 78.62 | 64.80 | 62.72 |
| | | SliceGPT | 45.48 | 73.53 | 69.42 | 45.00 | 75.35 | 70.80 | 63.26 |
| | | LaCo | 44.62 | 70.50 | 73.97 | 41.40 | 76.06 | 70.48 | 62.84 |
| | | LF (Avg) | **46.33** | _71.63_ | _74.71_ | 44.00 | _76.66_ | 70.17 | _63.92_ |
| | | LF (First) | 46.12 | 71.42 | **75.24** | **44.40** | 76.24 | **70.72** | 64.02 |
| | | LF (Centroid) | _46.25_ | **72.10** | 74.76 | 44.20 | **77.04** | _70.32_ | **64.11** |

testing models' overall capabilities in preserving core competencies and handling tasks of varying difficulty. This ensures our evaluation results remain comparable with mainstream research.

**LLMs** At the model level, we selected the most widely used Llama series models, specifically choosing **Llama 3.1-8B** Dubey et al. (2024) and **Llama 2-13B** Touvron et al. (2023), two models with different parameter counts—to demonstrate the stability of the proposed method.

### 4.2 PROBE DATA

The probe data is carefully constructed to cover diverse functional aspects of language model capabilities. We design samples spanning **multiple linguistic dimensions**, including syntactic processing, semantic comprehension, logical reasoning, knowledge retrieval, and mathematical computation. Each dimension contains representative examples that can effectively trigger different layers of the model to exhibit their specialized processing patterns. This multifaceted probe design ensures that our layer similarity analysis captures the true functional redundancy across various language understanding tasks. In all experiments, the total number of probe data points was set to **50**. The complete taxonomy and specific examples for each functional dimension are detailed in Appendix F.

### 4.3 MAIN RESULT

**Setup** We conducted an exhaustive performance comparison with existing mainstream model compression methods (without post-training). Specifically, our selected baseline methods include LLM-Pruner, SliceGPT, and LaCo. **LLM-Pruner** Ma et al. (2023) employs a structured pruning strategy that first divides the entire model into distinct subnetworks by identifying path dependencies, then selects which subnetworks to prune by estimating the importance of model weights. **SliceGPT** Ashkboos et al. (2024) uses orthogonal transformations to prioritize important dimensions of the input matrix, subsequently pruning the weight matrices of other dimensions to preserve the model's original functionality as much as possible. **LaCo** is a widely adopted layer-wise pruning method. It uses the model's first layer as a baseline, calculates the differences between subsequent layers

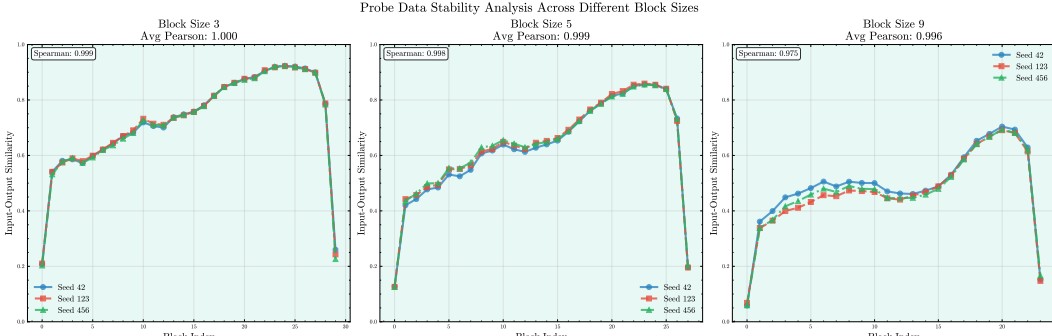

Figure 4: We selected **10** samples from a large probe database using different random seeds and employed these samples to compute input-output similarity, thereby verifying the stability of our method for probe data.

and the first layer, and directly adds these differences to the first layer to perform layer pruning. In practice, we observed that directly adding the differences caused the first layer's weight magnitudes to become excessively large, rendering the model ineffective. Therefore, we introduced an empirical coefficient at this step to control the magnitude of the differences and performed parameter tuning. **LF (Avg)**, **LF (First)**, and **LF (Centroid)** represent three methods for calculating the baseline weight: using the average of the inner layer weights within the block to be compressed, the first layer weights within the block, and the functionally weighted average of the inner layer weights within the block. These can be regarded as three variants of our method. The experimental results are shown in Table 1. More detailed settings are described in Appendix B.

**Layer Compression Vs. Structured Pruning** First, we can observe from the comparison between layer compression methods (LaCo and LF) and traditional structured pruning approaches (LLM-Pruner, SliceGPT) that layer compression consistently outperforms structured pruning. This suggests that structured pruning is more prone to disrupting the model's internal architecture, whereas layer compression, operating at a coarser granularity, avoids this issue and thus better preserves the model's capabilities.

**LaCo Vs. LF** Next, we focus on the internal comparison between the LaCo method and our LF method. It can be observed that under the Llama 3.1-8B model with a compression ratio of $25\%$, the average performance of LF surpasses that of LaCo by $14.81\%$. This indicates that our approach better leverages model information compared to LaCo in smaller models and under high compression rates, mitigating information loss caused by noise introduction during layer fusion. Although this difference tends to decrease as the model depth increases and the compression ratio decreases, our LF method consistently outperforms the LaCo method.

**Different Baseline Weight Calculation Strategies** Finally, by comparing several variants of the LF method, we observe that using functional weighting as the baseline weight consistently yields the best performance (highest Average metric) across different compression ratios and model depths. Under the Llama 3.1-8B model with 25% compression, it outperforms the LF(Avg) and LF(First) methods on the Average metric. This demonstrates that when the model is smaller (each layer is relatively more important) and the compression ratio is high, the functional weighting-derived baseline weight effectively guides the residual extraction and fusion process, leading to improved results. This can also be viewed as an **ablation study** of our LF method.

### 4.4 PROBE DATA SENSITIVITY

**Random Probe Data Selection** To further demonstrate the exceptional robustness of our method even under extremely sparse probe data conditions, we selected 10 probe data points from a large probe database (2000 entries) using three random seeds (42, 123, 456) to compute input-output similarity for layer compression. The results obtained under different block size settings are shown in Figure 4. It can be observed that despite the extremely limited quantity (only 10) and high randomness of the probe data, the similarity trends between different blocks are nearly consistent (have a **Pearson** correlation coefficient close to 1). Although absolute similarity values differ for blocks with greater lengths, this does not affect the similarity of the overall trend. The previous

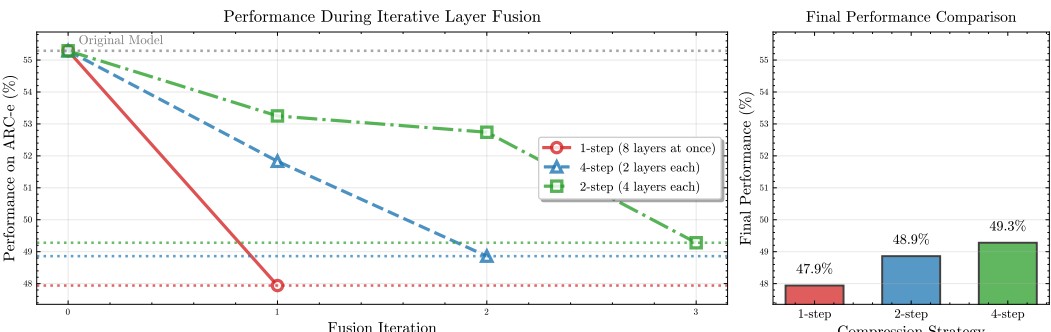

Figure 5: Visualization of Iterative Layer Fusion Effects for Different Step Lengths.

work Chen et al. (2024) included similar tests, but they only examined trends in scenarios with large datasets, whereas our experiments focus on scenarios with extremely small datasets.

### 4.5 ITERATIVE LAYER FUSION

**Setup** Unlike one-time direct model compression, our **LF** framework also holds significant potential for iterative layer fusion. Specifically, if we aim to achieve a high compression ratio $\rho$, we can perform multiple-layer fusions on a model using smaller compression ratios. This differs from direct layer fusion in that iterative fusion may utilize the weights from the previous fusion layer, thereby minimizing changes to the original model layers while ensuring the desired compression ratio. We employed Llama 3.1-8B with a 25% compression ratio (reducing 8 layers) as our iterative experimental setup. We selected three compression approaches: the first directly fused 8 layers in one step (**1 Step**), the second reduced 4 layers per iteration with 2 iterations (**2 Step**), and the third reduced 2 layers per iteration with 4 iterations (**4 Step**). The model's performance on ARC-e served as our evaluation metric. More detailed experimental settings are provided in Appendix G.

**Results Analysis** As shown in Figure 5, when maintaining the same compression rate, the multi-step iterative fusion method ultimately yields a model with superior performance. Moreover, as the number of iterations increases (with a corresponding decrease in the number of fusion layers per iteration), the compressed model demonstrates enhanced capabilities. Compared to single-step compression, the four-step iterative compression approach achieves approximately 1.4% higher performance. This demonstrates that the Layer Fusion method holds greater potential.

## 5 LIMITATION & FUTURE WORK

Current LF methods still face several issues that need to be addressed. Firstly, although the computation of baseline weights has been improved to some extent through strategies such as functional strength-weighted averaging, there remains a lack of intuitive analysis from perspectives such as loss landscapes. Moreover, more advanced methods for calculating baseline weights—such as those based on training dynamics, loss landscape properties, or requiring fewer hyperparameters—warrant further investigation in the future. Secondly, traditional model fusion methods typically treat layers within networks as parallel and independent entities, neglecting potential inter-layer dependencies. However, from the perspective of layer fusion, the weight of a subsequent layer often heavily depends on that of the preceding one. Thus, incorporating such hierarchical dependencies into the weight fusion process represents a promising research direction.

## 6 CONCLUSION

In this paper, we propose a **Layer Fusion** framework for large language models (LLMs). This method fuses consecutive layer weights into a single layer through specific steps, aiming to maximize the utilization of information contained within the layer weights to enhance the performance of compressed models. Simultaneously, our approach bridges the traditional **model fusion** domain with the **unstructured pruning** domain. While addressing the long-standing lack of practical applications in these fields, it also injects new vitality into the domain of LLM streamlining.

## 7 ETHICS STATEMENT

The research presented in this paper is based entirely on the analysis of existing, publicly available data. As no human or animal subjects were involved, and the data utilized are anonymous and do not contain any identifiable personal information, this study is exempt from ethical approval requirements.

## 8 REPRODUCIBILITY STATEMENT

We provide the main code files via **anonymous links** in the Abstract section of the paper, and include all hyperparameter settings potentially used in the experiments in the **Appendix**. We believe these details will help the community better understand our paper.

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

## A  LLM USAGE STATEMENT

Large Language Models (LLMs) were used to aid in the writing and polishing of the manuscript. Specifically, we used an LLM to assist in refining the language, improving readability, and ensuring clarity in various sections of the paper. The model helped with tasks such as sentence rephrasing, grammar checking, and enhancing the overall flow of the text.

It is important to note that the LLM was not involved in the ideation, research methodology, or experimental design. All research concepts, ideas, and analyses were developed and conducted by the authors. The contributions of the LLM were solely focused on improving the linguistic quality of the paper, with no involvement in the scientific content or data analysis.

The authors take full responsibility for the content of the manuscript, including any text generated or polished by the LLM. We have ensured that the LLM-generated text adheres to ethical guidelines and does not contribute to plagiarism or scientific misconduct.

## B  MAIN RESULT PARAMETER SETTINGS

In our experiments, we uniformly employed $50$ probe data inputs to extract input-output data for each model layer. These data points were evenly sampled from different probe data classifications. For both SliceGPT [1] and LLM-Pruner [2], we utilized their official open-source code repositories from GitHub for implementation. For the actual implementation of LaCo, we set the fusion coefficient to $0.2$ when fusing 8 layers. Since our **LF** method employs different granularities during fusion (8 layers, 4 layers, 2 layers), we set the fusion coefficients $\lambda$ to $(0.2, 0.4, 0.6)$ respectively. Additionally, during the Balancing phase, we measure the importance of each position weight. We set the **LV**s with importance in the bottom 80% to zero to enhance the fusion effect.

## C  MATHEMATICAL PROOF OF BASELINE WEIGHT SELECTION

In this section, we provide a mathematical proof demonstrating the superiority of using the mean of layer weights as the reference point over using the first layer's weights in our Layer Fusion framework.

### C.1  NOTATION AND DEFINITIONS

Let $\Theta = \{\theta_1, \theta_2, \ldots, \theta_N\}$ represent the weights of $N$ consecutive decoder layers, where each $\theta_i \in \mathbb{R}^d$ is a weight vector (or flattened weight matrix). We define the mean weight vector as:

$$\bar{\theta} = \frac{1}{N} \sum_{i=1}^{N} \theta_i$$

We consider two schemes for selecting the reference point $B$:

1. **Scheme A**: $B = \theta_1$ (First layer as reference)
2. **Scheme B**: $B = \bar{\theta}$ (Mean as reference)

The layer vector (LV) for each layer $i$ is defined as $LV_i = \theta_i - B$.

### C.2  MEAN PROPERTIES OF LAYER VECTORS

**Lemma 1.** *The mean of the layer vectors across all layers is zero for Scheme B but not necessarily for Scheme A.*

*Proof.* For Scheme A, the mean layer vector is:

$$\mu^{(A)} = \frac{1}{N} \sum_{i=1}^{N} LV_i^{(A)} = \frac{1}{N} \sum_{i=1}^{N} (\theta_i - \theta_1) = \bar{\theta} - \theta_1$$

which is generally non-zero unless $\theta_1 = \bar{\theta}$. For Scheme B, the mean layer vector is:

$$\mu^{(B)} = \frac{1}{N} \sum_{i=1}^{N} LV_i^{(B)} = \frac{1}{N} \sum_{i=1}^{N} (\theta_i - \bar{\theta}) = \bar{\theta} - \bar{\theta} = 0$$

$\square$

---

[1]https://github.com/microsoft/TransformerCompression
[2]https://github.com/horseee/LLM-Pruner

### C.3 Variance Analysis for Importance Measurement

In the balancing module, we measure the importance of each position $j$ in the weight vector by computing the variance of the layer vectors across layers:

$$\sigma_j^2 = \frac{1}{N} \sum_{i=1}^{N} (LV_{i,j} - \mu_j)^2$$

where $LV_{i,j}$ is the $j$-th element of $LV_i$ and $\mu_j$ is the $j$-th element of the mean layer vector.

**Theorem 1.** *The variance calculation in Scheme A introduces a systematic bias term $(\bar{\theta}_j - \theta_{1,j})^2$ that does not represent true inter-layer variation.*

*Proof.* For Scheme A, the variance at position $j$ is:

$$\sigma_j^{2(A)} = \frac{1}{N} \sum_{i=1}^{N} (LV_{i,j}^{(A)} - \mu_j^{(A)})^2$$

$$= \frac{1}{N} \sum_{i=1}^{N} [(\theta_{i,j} - \theta_{1,j}) - (\bar{\theta}_j - \theta_{1,j})]^2$$

$$= \frac{1}{N} \sum_{i=1}^{N} (\theta_{i,j} - \bar{\theta}_j)^2$$

However, the balancing module typically uses the second moment rather than the variance for importance measurement:

$$E_j^{(A)} = \frac{1}{N} \sum_{i=1}^{N} (LV_{i,j}^{(A)})^2$$

$$= \frac{1}{N} \sum_{i=1}^{N} (\theta_{i,j} - \theta_{1,j})^2$$

$$= \frac{1}{N} \sum_{i=1}^{N} [(\theta_{i,j} - \bar{\theta}_j) + (\bar{\theta}_j - \theta_{1,j})]^2$$

$$= \frac{1}{N} \sum_{i=1}^{N} (\theta_{i,j} - \bar{\theta}_j)^2 + (\bar{\theta}_j - \theta_{1,j})^2 + \frac{2}{N}(\bar{\theta}_j - \theta_{1,j}) \sum_{i=1}^{N} (\theta_{i,j} - \bar{\theta}_j)$$

$$= \sigma_j^{2(B)} + (\bar{\theta}_j - \theta_{1,j})^2$$

where $\sigma_j^{2(B)} = \frac{1}{N} \sum_{i=1}^{N} (\theta_{i,j} - \bar{\theta}_j)^2$ is the variance for Scheme B. The cross term vanishes because $\sum_{i=1}^{N} (\theta_{i,j} - \bar{\theta}_j) = 0$. □

The proof shows that Scheme B (using the mean as reference) provides a pure measurement of inter-layer variation in the balancing module, while Scheme A (using the first layer as reference) introduces a systematic bias term $(\bar{\theta}_j - \theta_{1,j})^2$. This bias term reflects the deviation of the first layer from the mean rather than true variation across layers, potentially leading to inaccurate importance measurements. Therefore, using the mean weight as the reference point is mathematically superior for the Layer Fusion framework, as it ensures that the balancing module can accurately identify important positions based solely on genuine inter-layer variation.

## D Proof of Functional Weighted Average Superiority

In this section, we provide a mathematical proof demonstrating the superiority of using the functional weighted average as the reference point over the simple average in the Layer Fusion framework. The proof hinges on the functional strength metric, which quantifies the influence of each layer on the hidden states. We show that the functional weighted average ensures a more accurate importance measurement in the balancing module, leading to better fusion results.

## D.1 NOTATION AND DEFINITIONS

Consider a block of consecutive decoder layers $S = \{\mathcal{L}_t, \mathcal{L}_{t+1}, \ldots, \mathcal{L}_{t+B-1}\}$ with $B$ layers. Each layer $\mathcal{L}_i$ has a weight vector $\theta_i \in \mathbb{R}^d$. The functional strength of a layer $\mathcal{L}_i$ is defined as:

$$\text{Func}(\mathcal{L}_i) = 1 - \text{Sim}(\mathcal{L}_i)$$

where $\text{Sim}(\mathcal{L}_i)$ is the cosine similarity between the input and output hidden states of $\mathcal{L}_i$. The functional strength is non-negative and higher values indicate stronger functionality. The weights for the weighted average are defined as:

$$w_i = \frac{\text{Func}(\mathcal{L}_i)}{\sum_{j=t}^{t+B-1} \text{Func}(\mathcal{L}_j)}, \quad \text{so that} \quad w_i \geq 0 \quad \text{and} \quad \sum_{i=t}^{t+B-1} w_i = 1.$$

We compare two schemes for selecting the reference point $B$:

1. **Scheme A (Simple Average):** $B_{\text{avg}} = \bar{\theta} = \frac{1}{B} \sum_{i=t}^{t+B-1} \theta_i$

2. **Scheme B (Functional Weighted Average):** $B_w = \theta_w = \sum_{i=t}^{t+B-1} w_i \theta_i$

The layer vector (LV) for each layer $i$ is defined as the deviation from the reference point:

$$LV_i = \theta_i - B$$

## D.2 WEIGHTED MEAN OF LAYER VECTORS

**Lemma 2.** *In Scheme B, the weighted mean of the layer vectors with weights $w_i$ is zero.*

*Proof.* The weighted mean of the layer vectors is:

$$\sum_{i=t}^{t+B-1} w_i LV_i^w = \sum_{i=t}^{t+B-1} w_i(\theta_i - \theta_w) = \sum_{i=t}^{t+B-1} w_i \theta_i - \theta_w \sum_{i=t}^{t+B-1} w_i = \theta_w - \theta_w = 0.$$

This shows that the weighted mean is zero, which is a desirable property for unbiased variance calculation in the balancing module. $\square$

In Scheme A, the simple mean of the layer vectors is:

$$\frac{1}{B} \sum_{i=t}^{t+B-1} LV_i^{\text{avg}} = \frac{1}{B} \sum_{i=t}^{t+B-1} (\theta_i - \bar{\theta}) = \bar{\theta} - \bar{\theta} = 0.$$

Thus, both schemes have a zero mean for the layer vectors under their respective weighting. However, the key difference lies in the variance calculation.

## D.3 VARIANCE CALCULATION FOR IMPORTANCE MEASUREMENT

In the balancing module, the importance of each position $j$ in the weight vector is measured by the variance of the layer vectors across layers. For Scheme B, we use the weighted variance:

$$\sigma_j^{2(w)} = \sum_{i=t}^{t+B-1} w_i(LV_{i,j}^w)^2$$

where $LV_{i,j}^w$ is the $j$-th element of $LV_i^w$. Since the weighted mean is zero, this is an unbiased estimator of the weighted variance. For Scheme A, the variance is computed with equal weights:

$$\sigma_j^{2(\text{avg})} = \frac{1}{B} \sum_{i=t}^{t+B-1} (LV_{i,j}^{\text{avg}})^2$$

To compare these variances, we model each weight vector as:

$$\theta_i = \theta^* + \epsilon_i$$

where $\theta^*$ is a common underlying weight vector, and $\epsilon_i$ is a layer-specific deviation capturing the functional characteristics. Layers with high functional strength have larger deviations $\epsilon_i$, meaning they exert more influence on the hidden states.

**Theorem 2.** *The weighted variance $\sigma_j^{2(w)}$ in Scheme B more accurately reflects the variability of functionally strong layers compared to the simple variance $\sigma_j^{2(avg)}$ in Scheme A.*

*Proof.* In Scheme B, the reference point is:

$$\theta_w = \sum_{i=t}^{t+B-1} w_i \theta_i = \theta^* + \sum_{i=t}^{t+B-1} w_i \epsilon_i.$$

Since $w_i$ is proportional to $\text{Func}(\mathcal{L}_i)$, and functionally strong layers have larger $\epsilon_i$, $\theta_w$ is biased towards these layers. The layer vector is:

$$LV_i^w = \theta_i - \theta_w = \epsilon_i - \sum_{k=t}^{t+B-1} w_k \epsilon_k.$$

The weighted variance at position $j$ is:

$$\sigma_j^{2(w)} = \sum_{i=t}^{t+B-1} w_i \left( \epsilon_{i,j} - \sum_{k=t}^{t+B-1} w_k \epsilon_{k,j} \right)^2.$$

This expression gives more weight to layers with large $w_i$ (i.e., functionally strong layers), so the variance is dominated by these layers. In Scheme A, the reference point is:

$$\bar{\theta} = \theta^* + \frac{1}{B} \sum_{i=t}^{t+B-1} \epsilon_i.$$

The layer vector is:

$$LV_i^{\text{avg}} = \epsilon_i - \frac{1}{B} \sum_{k=t}^{t+B-1} \epsilon_k.$$

The simple variance at position $j$ is:

$$\sigma_j^{2(\text{avg})} = \frac{1}{B} \sum_{i=t}^{t+B-1} \left( \epsilon_{i,j} - \frac{1}{B} \sum_{k=t}^{t+B-1} \epsilon_{k,j} \right)^2.$$

This variance treats all layers equally, regardless of their functional strength. Consequently, it may be unduly influenced by layers with low functional strength (small $\epsilon_i$), which act as noise, while diluting the signal from functionally strong layers. To quantify the difference, consider the expectation of the variances. Assume that the deviations $\epsilon_i$ are uncorrelated with mean zero and variance $\sigma_i^2$ for each layer. Then, for Scheme A:

$$\mathbb{E}[\sigma_j^{2(\text{avg})}] = \frac{1}{B} \sum_{i=t}^{t+B-1} \sigma_i^2 - \frac{1}{B^2} \sum_{i=t}^{t+B-1} \sigma_i^2 = \frac{B-1}{B^2} \sum_{i=t}^{t+B-1} \sigma_i^2.$$

For Scheme B, the weighted variance has expectation:

$$\mathbb{E}[\sigma_j^{2(w)}] = \sum_{i=t}^{t+B-1} w_i \sigma_i^2 - \sum_{i=t}^{t+B-1} w_i^2 \sigma_i^2.$$

Since $w_i$ is larger for layers with high functional strength (and thus large $\sigma_i^2$), Scheme B emphasizes layers with high variability. In contrast, Scheme A averages over all layers equally, which may suppress the signal from strong layers if there are many weak layers. Therefore, Scheme B provides a more accurate importance measurement by focusing on functionally strong layers, which are critical for fusion. □

### D.4 Conclusion

The functional weighted average scheme (Scheme B) is mathematically superior to the simple average scheme (Scheme A) because:

1. It ensures the weighted mean of layer vectors is zero, facilitating unbiased variance calculation.

2. It assigns higher weights to layers with strong functionality, so the variance calculation emphasizes these layers, leading to more accurate importance measurements in the balancing module.

3. It reduces the influence of layers with low functionality, which often contribute noise rather than signal.

This proof justifies the use of a functional weighted average in the Layer Fusion framework, as it enhances the quality of the fused weights by preserving the characteristics of functionally strong layers.

## E  Calculation Details for the Balancing Stage

In this stage, we perform feature-aware refinement of the extracted layer vectors through a mechanism inspired by Du et al. (2024). For a selected block $S = \{\mathcal{L}_t, \mathcal{L}_{t+1}, \ldots, \mathcal{L}_{t+B-1}\}$ with corresponding layer vectors $\{\mathbf{v}_t, \mathbf{v}_{t+1}, \ldots, \mathbf{v}_{t+B-1}\}$, we apply a balancing function $\psi : \mathbb{R}^d \to \mathbb{R}^d$ that adaptively reweights each dimension based on its significance.

The balancing process consists of two complementary components:

Intra-Balancing: Measures the importance of each parameter within a single layer vector. We compute a self-aware importance score vector $\boldsymbol{\beta}_{\text{intra},i} \in \mathbb{R}^d$ for each $\mathbf{v}_i$ using a normalized activation function (e.g., softmax) over squared magnitudes:

$$\boldsymbol{\beta}_{\text{intra},i} = \text{Softmax}\left(B \cdot \text{Norm}(|\mathbf{v}_i|^2)\right) \tag{18}$$

where the factor $B$ (number of layers in the block) regulates the degree of suppression applied to redundant parameters.

Inter-Balancing: Assesses cross-layer interactions by evaluating pairwise similarities between parameters at the same position across different layer vectors in the block. For the $k$-th parameter across all layer vectors, we compute:

$$\beta_{\text{inter},i}^{(k)} = \sum_{j=t}^{t+B-1} \text{Softmax}\left(\mathbf{v}_i^{(k)} \cdot \mathbf{v}_j^{(k)}\right) \tag{19}$$

which promotes consistency among important features and suppresses conflicting signals.

The final balanced layer vector is obtained via element-wise multiplication of the intra- and inter-balancing components:

$$\tilde{\mathbf{v}}_i = \psi(\mathbf{v}_i) = \mathbf{v}_i \odot \left(\boldsymbol{\beta}_{\text{intra},i} \odot \boldsymbol{\beta}_{\text{inter},i}\right).$$

This process enhances the representational quality of layer vectors by emphasizing informative features and reducing noise, thereby facilitating more effective fusion in the subsequent stage.

## F  Details for Probe Data

### F.1  Probe Data Structure Design

This study designs a structured probe data system to comprehensively evaluate language model performance across different linguistic functional dimensions. Each probe sample contains four core elements: probe text content, main category, specific subcategory, and expected language capability to be tested. The data structure is defined using a Python dataclass with fields for text content, category classification, subcategory specification, and expected capability assessment.

This structured design ensures the systematic nature and interpretability of probe data, facilitating subsequent analysis of how different task types affect model layers. The categorization scheme covers five main dimensions: syntax, semantics, reasoning, mathematics, and knowledge, each with specific subcategories to target particular linguistic capabilities.

## F.2 PROBE DATA ACQUISITION PROCESS

In the layer fusion system, probe data acquisition follows a systematic four-stage process:

**Probe Generation Phase:** A specialized probe generator creates a diverse set of probe samples, with the number of samples being configurable. The text content is then extracted from the generated probe set for subsequent processing.

**Text Encoding Phase:** The extracted probe texts are encoded using a tokenizer that converts them into tensor representations. This process includes padding and truncation operations to ensure uniform input length, with a specified maximum sequence length parameter.

**Embedding Representation Acquisition:** The encoded token IDs are transformed into embedding vectors using the model's embedding layer. This operation is performed in inference mode without gradient computation to preserve computational efficiency.

**Layer Activation Collection:** For each layer in the model, the input and output activation states are collected. The activations from the last token position are extracted, detached from the computation graph, and transferred to CPU memory for subsequent analysis. This process creates a comprehensive record of layer-wise information processing.

## F.3 REPRESENTATIVE PROBE SAMPLE DEMONSTRATION

Based on the comprehensive probe dataset, we selected representative samples organized in a structured table format:

## F.4 APPLICATION OF PROBE DATA IN LAYER FUSION

### INTER-LAYER SIMILARITY CALCULATION

The probe data enables calculation of input-output similarity for continuous layer blocks. The similarity computation involves extracting the input state from the starting layer and the output state from the ending layer of the block. Cosine similarity is then computed between corresponding input and output vectors across all samples in the batch. The final similarity metric represents the mean cosine similarity across all probe samples, providing a quantitative measure of information preservation through the layer block.

### FUNCTIONAL WEIGHT CALCULATION

Based on probe data, functional weights are computed for each layer to guide the fusion process. The calculation involves determining the input-output similarity for individual layers, where lower similarity indicates stronger functional transformation. The functional score is defined as one minus the layer similarity value. These scores are then normalized to create a probability distribution that reflects the relative functional importance of each layer within the specified range. The normalized scores are subsequently applied to weight the fusion process, emphasizing layers with stronger functional characteristics.

## F.5 QUALITY ASSURANCE OF PROBE DATA

To ensure the scientific validity and reliability of probe data, multiple quality assurance measures are implemented. Diversity is guaranteed through systematic sampling that uniformly distributes samples across all categories, preventing over-representation of any particular type while maintaining a configurable total sample size. Reproducibility is emphasized by storing probe data in a standardized JSON format, enabling exact experiment replication. Consistent data structure definitions ensure cross-experiment comparability, and the system supports loading and utilization of custom probe datasets for extended research applications.

Table 2: Representative Probe Samples Organized by Linguistic Category. Each sample is designed to test specific language understanding capabilities across **five major categories**: syntax, semantics, reasoning, mathematics, and knowledge.

| Probe Text | Category | Subcategory |
|---|---|---|
| **Syntax Understanding** | | |
| *Birds fly in the blue sky.* | **Syntax** | Simple Sentence |
| *Although it was raining, they continued their journey.* | **Syntax** | Complex Sentence |
| *Why did you choose this option?* | **Syntax** | Question Form |
| **Semantic Analysis** | | |
| *Apples and oranges are both fruits.* | **Semantics** | Word Relations |
| *Time is money.* | **Semantics** | Metaphor |
| **Logical Reasoning** | | |
| *Heavy objects fall down due to gravity.* | **Reasoning** | Common Sense |
| *All birds have wings. Penguins are birds. Therefore, penguins have wings.* | **Reasoning** | Logical Inference |
| **Mathematical Computation** | | |
| *$12 \times 4 = 48$* | **Mathematics** | Arithmetic |
| *A car travels 60 km/h. How far does it travel in 3 hours?* | **Mathematics** | Word Problems |
| **Factual Knowledge** | | |
| *The capital of France is Paris.* | **Knowledge** | Factual |
| *Water has the chemical formula $H_2O$.* | **Knowledge** | Scientific |

Through this systematic probe data construction methodology, we achieve a comprehensive and objective evaluation of functional characteristics across language model layers. The carefully designed probe system provides a reliable data foundation for informed layer fusion decisions, supporting robust experimental analysis and conclusive validation of research findings.

## G   DETAILS FOR ITERATIVE LAYER FUSION

In the iterative layer fusion experiment, we repeatedly ran our compression code. For the goal of compressing 8 layers, we performed two compressions using a block size of 4 and four compressions using a block size of 2. Each compression ran independently, meaning that layers obtained from previous fusion steps could be re-fused as ordinary layers. This approach preserves more original information compared to a single fusion step involving fewer layer weights, potentially leading to improved model performance. In this experiment, when multiple compression steps are involved, we retain the model after each compression step for evaluation.

