# OpenReview forum: "From Model Fusion to Layer Fusion: A Weight Perspective"
_ICLR.cc/2026/Conference — ICLR 2026 Conference Withdrawn Submission_

### Official Review · Reviewer_WKp6 · 2025-10-27

**Soundness:** 2
**Presentation:** 3
**Contribution:** 2
**Rating:** 4
**Confidence:** 4

**Summary:**

This paper proposes a novel Layer Fusion (LF) framework for compressing large language models (LLMs) by fusing consecutive Transformer layers into a single composite layer. The method consists of five key stages: identifying layer features using minimal probe data, deciding which layers to fuse based on functional similarity, extracting residual weights from a baseline, balancing parameter importance to reduce noise, and fusing the residuals to generate new weights. Unlike traditional compression techniques like pruning or distillation, LF requires no fine-tuning, uses very little data (≤50 samples), and preserves the original model structure, making it hardware-friendly. Experiments show that LF outperforms existing methods across multiple benchmarks and model architectures, achieving a better trade-off between performance and model size while demonstrating strong scalability and compatibility.

**Strengths:**

1.The method is highly efficient, as it requires no fine-tuning and relies on a very small set of probe data (≤50 samples), drastically reducing computational and data costs.

2.The performance of this method significantly surpasses that of LaCo, a layer merging approach based on simple averaging, across multiple models, with particularly notable advantages on the Llama3.1-8B model.

**Weaknesses:**

1.This paper omits several relevant baseline methods, such as ShortGPT[1], LLM-Streamline[2], and ReplaceMe[3], which are all straightforward layer pruning approaches. Among them, ShortGPT and ReplaceMe are training-free, while LLM-Streamline only involves training lightweight networks.

2.As direct comparisons with these layer pruning methods are not included, one can only refer to their original papers for context. For example, LLM-Streamline[2] reports a performance drop of approximately 10% when compressing around 25% of the parameters in Llama3.1-8B, while ReplaceMe[3] demonstrates comparable performance. These results appear notably better than the Llama3 results in Table 1, which leads to a question regarding whether the proposed layer merging strategy is indeed superior to these simpler layer pruning methods.

[1]Men, Xin, et al. "Shortgpt: Layers in large language models are more redundant than you expect." arXiv preprint arXiv:2403.03853 (2024).

[2]Chen, Xiaodong, et al. "Streamlining redundant layers to compress large language models." arXiv preprint arXiv:2403.19135 (2024).

[3]Shopkhoev, Dmitriy, et al. "ReplaceMe: Network Simplification via Layer Pruning and Linear Transformations." arXiv preprint arXiv:2505.02819 (2025).

**Questions:**

1.Is it possible to provide experimental results demonstrating that the proposed method outperforms training-free layer pruning methods, such as ReplaceMe?

---

### Official Review · Reviewer_V8Bn · 2025-11-01

**Soundness:** 2
**Presentation:** 2
**Contribution:** 2
**Rating:** 4
**Confidence:** 3

**Summary:**

The paper proposes a layer fusion method that fuses several contiguous layers into one by measuring functional strength, identifying low-strength blocks, computing a centroid weight, and extracting residual layer vectors. It targets zero-shot performance and demonstrates this on Llama 3.1-8B and Llama 2-13B. The proposed method outperforms previous structure pruning methods and layer fusion methods at compression ratios of 12.5% and 25% on several benchmarks.

**Strengths:**

- The 5-step workflow is easy to follow.
- It is training-free and only requires a few probe samples to estimate the block importance.
- Iterative fusion is a promising approach. The experiment demonstrates that four small fusions are more effective than one large fusion. It is a concrete empirical insight, and it actually supports their own method.
- Results on the evaluation benchmark outperform baselines.

**Weaknesses:**

- The paper didn't enumerate which per-layer parameters are included (Q/K/V/O, MLP, RMSNorm, etc.).
- The final fused layer index is not shown. It would be helpful to understand the redundancy of the base model.
- Comparison might be unfair. Baseline configuration may be unfair or under-tuned. For Llama-2-13B, at 25% SliceGPT scores are lower than the officially reported ones (ARC-c, ARC-e).
- Lack of comparison. ShortGPT[1] and SLEB[2] are not compared, which also study layer redundancy for LLMs.
- Lack of study on similarity functions. The paper adopts cosine similarity; however, cosine similarity can not reveal magnitude relations.

[1] Men, Xin, et al. "Shortgpt: Layers in large language models are more redundant than you expect." (2024)
[2] Song, Jiwon, et al. "SLEB: Streamlining LLMs through Redundancy Verification and Elimination of Transformer Blocks." ICML, 2024.

**Questions:**

- Eq.7 is confusing, the subscriptions remain the same across all blocks from $t-1 \rightarrow t+B-1$, but the superscription $(n)$ is also related to the block index.
- Can you provide 20% and 30% pruning ratio results? They are also common ratios in other baselines.
- Did you try different similarity functions?
- What about the method compared with ShortGPT and SLEB.

---

### Official Review · Reviewer_djvr · 2025-11-01

**Soundness:** 2
**Presentation:** 2
**Contribution:** 2
**Rating:** 4
**Confidence:** 4

**Summary:**

This paper introduces a Layer Fusion (LF) framework that compresses large language models by fusing the weights of multiple contiguous Transformer layers into a single composite layer. LF identifies functionally redundant layers based on input–output hidden-state similarity, computes residual “layer vectors,” balances their importance, and generates fused layers via a functional weighted average of parameters. Experiments on Llama 3.1-8B and Llama 2-13B across multiple reasoning and commonsense benchmarks (ARC, HellaSwag, PIQA, OpenBookQA, Winogrande) show that LF consistently outperforms structured-pruning methods (LLM-Pruner, SliceGPT) and the recent layer-collapse baseline LaCo.

**Strengths:**

1. The paper reframes layer pruning as a weight-fusion problem, connecting model-merging with structual compression.
2. No large datasets are required, only probing activations from no more than 50 samples.

**Weaknesses:**

1. Probe data sensitivity. The probe design (hand-crafted synthetic categories) may bias which redundancies are detected. It is uncertain how LF behaves when probe data distribution mismatches real downstream usage.

2. No efficiency metrics such as latency or FLOPs are presented.

3. Compression ratios beyond 25\% and 12.5\% are not explored.

4. No study on the similarity functions. Cosine similarity is only one choice, other functions such as MSE or correlation coefficiencts can also be possible choices.

**Questions:**

1. What is the efficiency gain (memory, latency, FLOPs) of the proposed method?
2. Ablation studies for probe size, fusion coefficient λ, and balancing threshold.
3. What is the influence of the choice of similarity functions? Have you tried other functions such as MSE?
4. What is the performance other than 25\% and 12.5\% pruning ratios?

---

### Official Review · Reviewer_45W9 · 2025-11-02

**Soundness:** 2
**Presentation:** 2
**Contribution:** 2
**Rating:** 2
**Confidence:** 4

**Summary:**

This paper proposes Layer Fusion (LF), a training-free framework that compresses large language models by merging the weights of adjacent Transformer layers instead of removing them. Using small probe data to measure layer similarity, LF identifies redundant layers, computes residuals, balances important parameters, and fuses them into composite layers. Experiments show LF achieves a better accuracy–efficiency trade-off than strong baselines.

**Strengths:**

The method proposed in this paper appears to show improvements compared to the baselines.

**Weaknesses:**

1. The title is unclear. Readers may assume the paper provides a theoretical explanation of layer fusion from a weight perspective, but in fact, it proposes a pruning-based method. Since layer pruning itself is not novel and is not first introduced in this work, the title fails to accurately capture the paper’s actual contribution.
2. The idea of layer fusion in this paper is not new and is quite similar to LaCo. The main difference is that this work more finely evaluates which layers can be merged with minimal performance loss. However, the block size (B) is a fixed hyperparameter, so the method cannot explore all possible values and is not globally optimal. Overall, it performs only slightly better than LaCo, with limited improvement except on LLaMA-3.1-8B.
3. Compared with the dense models, the proposed method still shows a significant performance drop. Although it performs better than the baselines, it cannot reasonably claim to be “no fine-tuning required.”
4. The paper should include experiments on more model families and larger model sizes to strengthen its empirical validation. It also needs post-training experiments to demonstrate the potential of the pruned models to recover performance.
5. The paper should compare against more baselines, such as ShortGPT.
6. In terms of writing, the paper uses excessive bold formatting, especially in the abstract, which is uncommon.
7. The citation format in this paper is incorrect; in many places, \citep should be used instead of \citet.

**Questions:**

see above

---

### Note · Authors · 2025-11-14

I have read and agree with the venue's withdrawal policy on behalf of myself and my co-authors.